# A Clean Air Plan for Sydney: An Overview of the Special Issue on Air Quality in New South Wales

**Clare Paton-Walsh** [1,2,*], **Peter Rayner** [3], **Jack Simmons** [1,2], **Sonya L. Fiddes** [3,4,5], **Robyn Schofield** [3], **Howard Bridgman** [6], **Stephanie Beaupark** [1,2], **Richard Broome** [7], **Scott D. Chambers** [1,8], **Lisa Tzu-Chi Chang** [9], **Martin Cope** [5], **Christine T. Cowie** [10,11,12], **Maximilien Desservettaz** [1], **Doreena Dominick** [1,2], **Kathryn Emmerson** [5], **Hugh Forehead** [13], **Ian E. Galbally** [1,5], **Alan Griffiths** [1,8], **Élise-Andrée Guérette** [1,5], **Alison Haynes** [2,14], **Jane Heyworth** [12,15], **Bin Jalaludin** [12,16], **Ruby Kan** [9], **Melita Keywood** [5], **Khalia Monk** [9], **Geoffrey G. Morgan** [12,17], **Hiep Nguyen Duc** [9], **Frances Phillips** [1,2], **Robert Popek** [18], **Yvonne Scorgie** [9], **Jeremy D. Silver** [3], **Steve Utembe** [3,19], **Imogen Wadlow** [1,3], **Stephen R. Wilson** [1,2] and **Yang Zhang** [20]

1. Centre for Atmospheric Chemistry, University of Wollongong, Wollongong, NSW 2522, Australia; js828@uowmail.edu.au (J.S.); beaupark@uow.edu.au (S.B.); szc@ansto.gov.au (S.D.C.); mjd232@uowmail.edu.au (M.D.); dd824@uowmail.edu.au (D.D.); Ian.Galbally@csiro.au (I.E.G.); agf@ansto.gov.au (A.G.); elise-andree.guerette@csiro.au (É.-A.G.); francesp@uow.edu.au (F.P.); imogenwadlow@gmail.com (I.W.); swilson@uow.edu.au (S.R.W.)
2. School of Earth, Atmospheric and Life Sciences, University of Wollongong, Wollongong, NSW 2522, Australia; alison.haynes@internode.on.net
3. School of Earth Sciences, University of Melbourne, Melbourne, VIC 3010, Australia; prayner@unimelb.edu.au (P.R.); sonya.fiddes@climate-energy-college.org (S.L.F.); robyn.schofield@unimelb.edu.au (R.S.); jeremy.silver@unimelb.edu.au (J.D.S.); steven.utembe@unimelb.edu.au (S.U.)
4. Climate and Energy College, University of Melbourne, Melbourne, VIC 3010, Australia
5. Climate Science Centre, CSIRO Oceans and Atmosphere, Aspendale, VIC 3195, Australia; Martin.Cope@csiro.au (M.C.); Kathryn.Emmerson@csiro.au (K.E.); Melita.Keywood@csiro.au (M.K.)
6. School of Environmental and Life Sciences, University of Newcastle, Callaghan, NSW 2308, Australia; howard.bridgman@newcastle.edu.au
7. Environmental Health Branch, Health Protection NSW, Sydney, NSW 2050, Australia; Richard.Broome@health.nsw.gov.au
8. Australian Nuclear Science and Technology Organisation (ANSTO), Lucas Heights, NSW 2234, Australia
9. New South Wales Department of Planning, Industry and Environment, Lidcombe, Sydney, NSW 2141, Australia; LisaTzu-Chi.Chang@environment.nsw.gov.au (L.T.-C.C.); Ruby.Kan@environment.nsw.gov.au (R.K.); Khalia.Monk@environment.nsw.gov.au (K.M.); Hiep.Duc@environment.nsw.gov.au (H.N.D.); Yvonne.Scorgie@environment.nsw.gov.au (Y.S.)
10. South West Sydney Clinical School, University of New South Wales & Ingham Institute of Medical Research, Liverpool, NSW 2170, Australia; christine.cowie@sydney.edu.au
11. Woolcock Institute of Medical Research, University of Sydney, Sydney, NSW 2037, Australia
12. Centre for Air Pollution, Energy and Health Research, Glebe, Sydney, NSW 2037, Australia; jane.heyworth@uwa.edu.au (J.H.); Bin.Jalaludin@health.nsw.gov.au (B.J.); geoffrey.morgan@sydney.edu.au (G.G.M.)
13. SMART Infrastructure Facility, University of Wollongong, Wollongong, NSW 2522, Australia; hughf@uow.edu.au
14. Centre for Sustainable Ecosystem Solutions, University of Wollongong, Wollongong, NSW 2522, Australia
15. School of Population and Global Health, University of Western Australia, Curtin, WA 6907, Australia
16. Ingham Institute for Applied Medical Research, University of New South Wales, Liverpool, NSW 2170, Australia
17. School of Public Health, Faculty Medicine and Health, University of Sydney, Sydney, NSW 2006, Australia
18. Department of Basic Research in Horticulture, Institute of Horticultural Sciences, Faculty of Horticulture and Biotechnology, Warsaw University of Life Sciences—SGGW, 02-787 Warsaw, Poland; robert.popek@gmail.com

19　Environment Protection Authority, Victoria, Melbourne, VIC 3001, Australia
20　Department of Marine, Earth and Atmospheric Sciences, North Carolina State University, Raleigh, NC 27695, USA; yzhang9@ncsu.edu
*　Correspondence: clarem@uow.edu.au

**Abstract:** This paper presents a summary of the key findings of the special issue of Atmosphere on Air Quality in New South Wales and discusses the implications of the work for policy makers and individuals. This special edition presents new air quality research in Australia undertaken by (or in association with) the Clean Air and Urban Landscapes hub, which is funded by the National Environmental Science Program on behalf of the Australian Government's Department of the Environment and Energy. Air pollution in Australian cities is generally low, with typical concentrations of key pollutants at much lower levels than experienced in comparable cities in many other parts of the world. Australian cities do experience occasional exceedances in ozone and $PM_{2.5}$ (above air pollution guidelines), as well as extreme pollution events, often as a result of bushfires, dust storms, or heatwaves. Even in the absence of extreme events, natural emissions play a significant role in influencing the Australian urban environment, due to the remoteness from large regional anthropogenic emission sources. By studying air quality in Australia, we can gain a greater understanding of the underlying atmospheric chemistry and health risks in less polluted atmospheric environments, and the health benefits of continued reduction in air pollution. These conditions may be representative of future air quality scenarios for parts of the Northern Hemisphere, as legislation and cleaner technologies reduce anthropogenic air pollution in European, American, and Asian cities. However, in many instances, current legislation regarding emissions in Australia is significantly more lax than in other developed countries, making Australia vulnerable to worsening air pollution in association with future population growth. The need to avoid complacency is highlighted by recent epidemiological research, reporting associations between air pollution and adverse health outcomes even at air pollutant concentrations that are lower than Australia's national air quality standards. Improving air quality is expected to improve health outcomes at any pollution level, with specific benefits projected for reductions in long-term exposure to average $PM_{2.5}$ concentrations.

**Keywords:** air quality; New South Wales; traffic; smoke; urban greening

---

## 1. Introduction

### 1.1. Objectives of This Review Paper

The special issue of Atmosphere on Air Quality in New South Wales, Australia, brings together papers that describe the outcomes of research undertaken by the Clean Air and Urban Landscapes (CAUL) hub and its collaborators, including a number of measurement campaigns, and a series of papers describing the results of the first major comparison of air quality models in Australia.

In this overview paper, we aim to:

1.　review the existing literature relevant for understanding air quality in New South Wales;
2.　summarise the key findings of research included in this special issue of *Atmosphere* (with an emphasis on the implications for policy makers); and
3.　finally, we outline a number of policy options that we believe should be prioritised, along with supporting evidence from this research and the wider scientific literature.

## 1.2. Air Quality in Sydney

Air pollution has recently been identified as the largest environmental risk factor to human health worldwide, with fine particulate matter the greatest contributor to impacts from poor air quality [1]. Sydney, in the state of New South Wales (NSW), is a city located on the south-eastern coast of Australia (33°52′ S, 151°12′ E). It has a population of 5.1 million people (as of June 2017) [2] and experiences a temperate climate with warm summers and no defined dry season [3]. Sydney has a similar latitude and continental position to the South American cities of Montevideo, Uruguay and Buenos Aires, Argentina and the South African city of Cape Town. Much of the urban area of the city of Sydney lies within a basin with elevated topography bounding the basin to the north, west, and south, with the Tasman Sea located at the eastern extent of the basin. The temperate coastal basin geography of Sydney means it is influenced by both synoptic and meso-scale meteorological phenomena [4]. In particular, cold air drainage into the basin during the cooler part of the year and afternoon sea breezes in the warmer months are frequent, persistent meso-scale processes that impact the city's air quality [5,6].

The New South Wales Department of Planning, Industry and Environment (DPIE) operates an extensive network of air quality monitoring stations across the Sydney region, monitoring six different measures of 'criteria air pollutants', which are used as indicators of air quality in NSW. These are ozone ($O_3$), nitrogen dioxide ($NO_2$), carbon monoxide (CO), sulfur dioxide ($SO_2$), visibility, and fine particles (including those with aerodynamic diameters below 10 microns and 2.5 microns known as $PM_{10}$ and $PM_{2.5}$, respectively). Measurements of each pollutant are normalised to the standard specified in the National Environment Pollution Measure for Ambient Air Quality (NEPM) [7]. This normalised information is published as the 'Air Quality Index' (AQI) for each pollutant. The air quality index for a monitoring station is determined by the highest criteria pollutant AQI. Similarly, the AQI for each region is determined by the highest station AQI [8].

Despite the relatively low air pollutant concentrations in Sydney in relation to comparable international cities, significant health impacts occur in the city [9]. Exposure (even at low concentrations) to pollutants, such as CO, nitrogen oxides ($NO_X$), and fine particulate matter (PM), has been found to increase hospital admissions for five outcomes of cardiovascular disease in elderly people in Sydney [10]. A longitudinal cohort study has found evidence of a detrimental association between long term exposure to the low concentrations of $PM_{2.5}$ and $NO_2$ in Sydney [11]. Table 1 lists the criteria pollutant standards, illustrating that some of Australia's NEPM standards are significantly less stringent than the levels recommended internationally, by either the World Health Organization or other developed nations.

**Table 1.** Table of Pollutant Standards.

| Criteria Pollutant | Australian NEPM Pollutant Standard * | World Health Organization Recommendation (ppb) or International if No WHO Recommendation |
|---|---|---|
| $SO_2$ 1 h | 200 ppb (100 ppb proposed) | European Union 124 ppb; USA 75 ppb (99th percentile of daily worst hour) |
| $SO_2$ 24 h | 80 ppb (20 ppb proposed) | 7.6 ppb |
| $SO_2$ annual | 20 ppb (proposed to be removed) | No standard |
| $NO_2$ 1 h | 120 ppb (90 ppb proposed) | 97 ppb |
| $NO_2$ annual | 30 ppb (19 ppb proposed) | 19 ppb |
| $O_3$ 1 h | 100 ppb (proposed to be removed) | New Zealand: 70 ppb; Japan: 60 ppb |
| $O_3$ 4 h | 80 ppb (proposed to be removed) | No standard |
| $O_3$ 8 h | No standard (65 ppb proposed) | 47 ppb |
| CO | 9 ppm | 10 ppm |
| $PM_{2.5}$ 24-h | 25 $\mu g/m^3$ (2025 goal of 20 $\mu g/m^3$) | 25 $\mu g/m^3$ |
| $PM_{2.5}$ annual | 8 $\mu g/m^3$ (2025 goal of 7 $\mu g/m^3$) | 10 $\mu g/m^3$ |
| $PM_{10}$ 24-h | 50 $\mu g/m^3$ | 50 $\mu g/m^3$ |
| $PM_{10}$ annual | 25 $\mu g/m^3$ | 20 $\mu g/m^3$ |

* Standards are currently being reviewed for $SO_2$, $NO_2$ and $O_3$—those proposed are those found in the consultation document [12].

Based on Australia's NEPM standards and the normalised AQI, between 2012 to 2018, 87% to 98% of days fell within the 'very good', 'good' or 'fair' category for Sydney, as illustrated in Figure 1. In recent years levels of $NO_2$, $SO_2$, and CO remained below their relevant standards in Sydney. Particle concentrations ($PM_{2.5}$ and $PM_{10}$) show year to year variations due to changes in weather, fires, and dust storms with no clear long-term trends apparent [13]. Relatively better air quality was recorded in wetter, cooler years (e.g., 2012) compared to the more recent hotter, drier years. In 2018, $PM_{2.5}$ and $PM_{10}$ levels were generally higher across the State due to impacts from intense drought conditions. In Sydney, $PM_{2.5}$ measurements were greater than the national standard (25 µg m$^{-3}$) on 19 days of 2018, while $PM_{10}$ concentrations exceeded the standard (50 µg m$^{-3}$) on 20 days. In fact, air quality was classified as 'hazardous' for 36 days of 2018 [14]. This classification was mostly associated with dust storms and hazard reduction burns or bushfires during autumn and winter, and during these episodes, the majority of Sydney's air quality monitoring stations recorded daily particle averages above the NEPM standard.

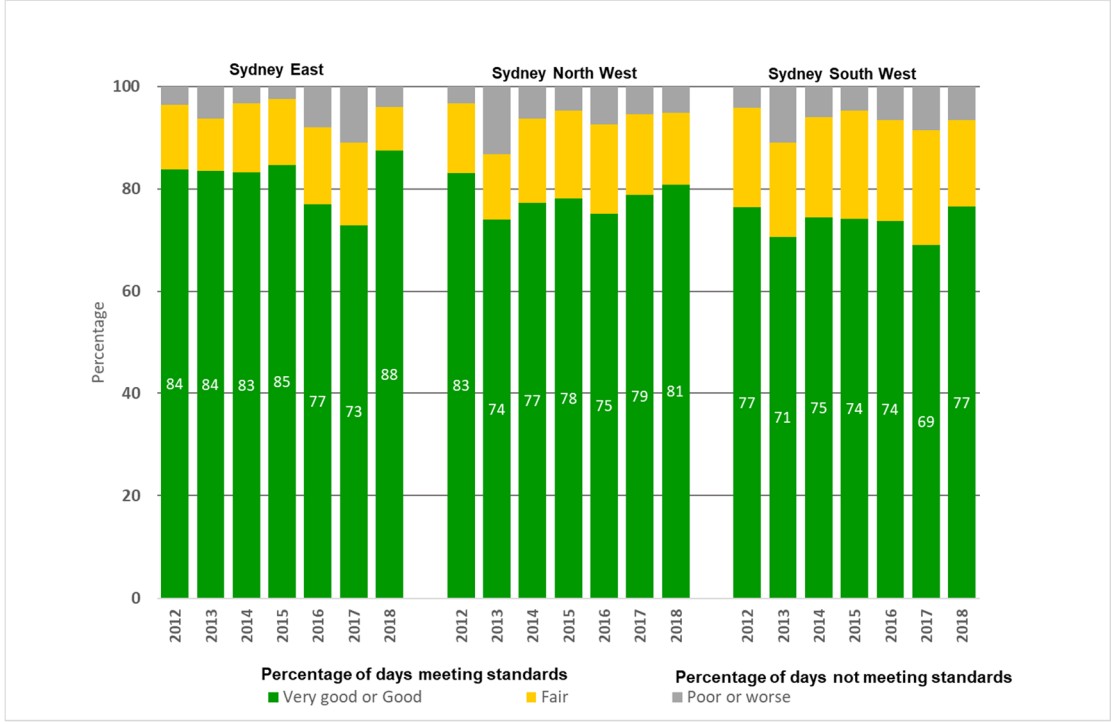

**Figure 1.** Percentage of days that the Air Quality Index (AQI) rated as 'very good' or 'good'; 'fair'; and 'poor' or worse, in different regions of Sydney between 2012 and 2018 [14].

$O_3$ was below the national standard for all but nine days in Sydney between 2015 and 2017 and seven days in 2018. Due to the fact that $O_3$ is photo-chemically produced from precursor pollutants, $O_3$ exceedances tend to be associated with high temperatures in the Sydney basin [14,15], which are already associated with adverse health outcomes [16]. Ozone exceedances are less common in Sydney's east where sea breezes are common in the afternoons and temperatures are lower in the summer, and this is reflected in the greater number of days with 'good' or 'very good' air quality (see Figure 1).

The NSW Environment Protection Authority creates 5-yearly emissions inventories for the greater metropolitan area (which includes Sydney and the neighbouring cities of Wollongong and Newcastle, which together account for 78% of the NSW population). The most recent emissions inventory was completed for the calendar year of 2008. The inventory for 2013 is in preparation. Anthropogenic contributions dominated total direct emissions of CO (97%), $PM_{10}$ (81%), $NO_X$ (98.3%), $PM_{2.5}$ (92%), and $SO_2$ (99%) [17]. More than 80% of $NO_X$ emissions (62% of $NO_2$) are attributed to on-road and off-road combustion engines. Approximately 70% of $PM_{10}$ emissions are from automobiles and

industrial sources, whereas more than 50% of $PM_{2.5}$ emissions arise from domestic-commercial sources. Care must be taken not to over-interpret these results because $O_3$ and secondary $PM_{2.5}$ are produced via atmospheric chemistry and therefore, indirect emissions must not be overlooked. Secondary $PM_{2.5}$ has been found to contribute significantly to total $PM_{2.5}$ concentrations in the Sydney basin [18,19]. More detailed source analysis is available for each source category on the NSW Environment Protection Authority's website [20]; however, these are estimates only since the large scale effort in compiling this inventory means that will be out of date by the time it is available.

Despite the limited number of recent air quality exceedances in Sydney, exposure to criteria pollutants below regulated standards are still associated with significant health effects [9,11,21,22]. Exposure to $PM_{2.5}$ concentrations considered as generally safe has been shown to increase the onset risk of stroke within hours [23]. In a cohort of older men in Perth, an increase in mortality was observed with increased exposure to $PM_{2.5}$ (measured as particle light absorbance, which is a surrogate for black carbon) [22]. Long term exposure to $PM_{2.5}$ concentrations in the greater Sydney region was associated with increased all-cause mortality in a study for the '45 and Up' cohort (people aged 45 years and older) [11]. These findings are consistent with international research, for example, significant adverse health effects due to exposure to $PM_{2.5}$ and $O_3$ concentrations below US EPA (United States Environmental Protection Agency) standards were found to increase all-cause mortality in a large population study of at-risk groups in the United States [24]. Due to the occasionally high levels of $PM_{2.5}$, and $O_3$ in Sydney, these pollutants will be discussed in depth below. Anthropogenic sources contributing significantly to air pollution exposures and episodic events impacting on Sydney's air quality such as fires, dust storms, and pollen will also be discussed.

*1.3. Particulate Pollution*

Typical levels of $PM_{2.5}$ pollution in both regional and urban Australia are generally low relative to comparable populated regions around the globe [25]. The concentration of $PM_{2.5}$ in Australian urban areas is modelled to have an annual average of 8 μg m$^{-3}$ [25]. New Zealand, a similarly developed nation, much smaller in land mass and population, and with less frequent forest fires, has an even lower annual average modelled concentration of 5 μg m$^{-3}$ [25]. Modelled annual average concentration of $PM_{2.5}$ in urban areas is lower in Australia than in nations with higher population densities and which experience substantial inter-regional transport of air pollution, such as the United Kingdom (12 μg m$^{-3}$), Germany (14 μg m$^{-3}$), and China (59 μg m$^{-3}$) [25]. The Australian annual average $PM_{2.5}$ standard (8 μg m$^{-3}$) is more stringent than standards or guideline values set by the European Union, United States, and the World Health Organization [26]. Nevertheless, on shorter time-scales, Sydney does experience exceedances of the daily $PM_{2.5}$ limit of 25 μg m$^{-3}$, and these exceedances are often associated with wildfires or hazard reduction burns [27,28].

There is some evidence that $PM_{2.5}$ concentrations are increasing in Sydney: a positive interannual trend was found in weekly mean $PM_{2.5}$ concentrations in all seasons at four Sydney sites [29]. An annual cycle in $PM_{2.5}$ monthly means was also found at the four sites examined in the study by Di Virgilio [29], with all sites displaying a May peak, high concentrations until early Austral summer before descending to a March minimum [29]. This is possibly due to the high incidence of hazard reduction burns during late autumn, and the consistent contribution of domestic wood heaters to particle concentrations in winter. Planetary boundary layer height and total cloud cover were the most consistent predictors of $PM_{2.5}$ concentration during high pollution days since hazard reduction burns are usually conducted on cool, clear days with light winds and low mixing heights [29].

No similar trend is found in $PM_{10}$ measurements: for example, Roberts [30] observed no change in $PM_{10}$ concentration from 1993–2007 in eastern Australian cities. This is unlike some cities in the eastern United States in which $PM_{10}$ concentrations are decreasing [31]. A synoptic climatology of the warm months found that elevated $PM_{10}$ concentrations in Sydney were associated with synoptic westerly winds [4], most likely carrying crustal dust over the city from the dry interior of NSW and from further inland areas.

A comprehensive study of $PM_{2.5}$ in Sydney, including both observational and modelling components, was undertaken during the Sydney Particle Study (SPS) [19,32–34]. Two month-long measurement campaigns were completed, in summer 2011 and autumn 2012, at Westmead Air Quality Station in the Sydney Basin. Summer measurements revealed strong contributions from sea salt (34%) and organic matter (34%) to the average $PM_{2.5}$ composition. The remainder of the particle fraction was attributed to secondary inorganic aerosol (15%), soil (11%) and elemental carbon (6%). In comparison, autumn particle loadings were dominated by organic matter (57%) with additional contributions from elemental carbon (16%), secondary inorganic aerosol (15%), soil (7%), and sea salt (5%). The elevated contribution of organic matter to the autumn particle loading is likely due to the use of domestic wood heating.

### 1.4. Ozone Pollution

Sydney experiences several $O_3$ exceedances each year, with high $O_3$ concentrations associated with high temperatures in the Sydney basin and more prevalent in the west of the city [14,15]. In contrast, background $O_3$ concentrations in Sydney are comparatively low. The annual mean ozone concentration for Sydney was 18.5 ppb in 2017 [35]. By comparison, the 2017 mean ozone concentration for urban sites in the UK was 27.9 ppb [36]. Previous measurements, taken over a decade ago, indicate that night-time background ozone concentration was in the range of 16–21 ppb at 14 air quality monitoring sites in the Sydney region [37] for the period 1998–2005. The daytime background oxidant ($O_3$ + $NO_2$) concentration in the Sydney region was about 35 ppb in 2005 for days in which no $O_3$ exceedances occurred, and 55 ppb for the ten days in which at least one site experienced $O_3$ concentrations above 80 ppb [38].

A positive trend in ozone concentrations has been measured in cities in Europe (e.g., London, Paris, and Berlin) and North America (e.g., Sacramento CA, Tucson AZ, and Dallas TX), linked to reductions in $NO_X$ pollution [39,40]. Though there is no consistent global trend [41–43], increases in background ozone concentrations have also been observed at pristine sites (e.g., Cape Grim, Tasmania, and Mauna Loa Hawaii) [44].

$O_3$ levels in Sydney increased from 1994 to 2002 (attributed to increasing temperatures, increasing regional anthropogenic emissions, and an increasing incidence of bushfires and hazard reduction burns) but have decreased again since 2002 with some annual variations [13,37]. Though ozone concentrations in Sydney are often relatively low, photochemical smog events have impacted the city in the past. $O_3$ concentration in Sydney is highest during the summer, particularly on very hot days. A smog event occurred in Sydney in January 2001, during which $O_3$ concentrations were measured at above 80 ppb for seven consecutive days [45]. A number of synoptic climatologies have been performed examining periods of unusually high $O_3$ concentration in the Sydney basin, and considering the impact of mesoscale air flows, such as basin drainage and sea breezes [4,46,47]. One of these studies created eleven synoptic categories for Sydney and found one synoptic category dominated during more than 90% of the days when daily maximum 1 h mean ozone concentrations were greater than 100 ppb [48]. This category was associated with an anticyclone in the central Tasman Sea coupled with a ridge to the northeast. A high inland-coast temperature gradient (leading to a significant sea breeze) and low mixing layer height were also characteristic. High $O_3$ concentrations were associated with a similar synoptic category [47]. However, while the synoptic categories were a reasonable predictor of high $O_3$ on average, exceedances occurred under almost all synoptic types. Therefore, a more directed examination was performed, focusing on the warm months of November to March [4]. Again, high pressure over the Tasman Sea, associated with high local temperatures and sea breezes was a predictor of high $O_3$ concentrations. Discrimination of exceedance events between synoptic categories was not significantly improved, highlighting the importance of mesoscale effects in the variation and the non-homogeneity of air pollutant concentrations across the basin.

Despite the difficulty in synoptically categorising ozone exceedances, most incidences of elevated ozone concentration experienced in Sydney are associated with a $NO_X$-limited regime. $O_3$ is formed

via the photochemical reaction of VOCs and $NO_X$ and is, therefore, limited by one of these reagents or by sunlight. The CSIRO Integrated Empirical Rate model was used to demonstrate that for all averaging periods, a $NO_X$-limited regime was present during most exceedance events (81%), especially in Western Sydney (92%) [49].

*1.5. Major Anthropogenic Sources of Air Pollution*

Although Sydney's most extreme pollution events are often associated with fires and dust storms (e.g., [28,50–52]), studies analysing the sources of $PM_{2.5}$ and $NO_X$ in NSW concluded that motor vehicle exhaust was the largest contributor to total $NO_X$ [53] and the largest anthropogenic contributor to total $PM_{2.5}$ [54]. Power stations are also implicated as a major source of atmospheric fine particles [18].

Road traffic makes a significant (5–80%) contribution to airborne concentrations of fine particulate matter (expressed both by mass and by number concentration) in urban areas around the world, with sources of particles from exhaust and non-exhaust (tyre wear, road dust, etc.) [55]. Exhaust emissions are dominant currently; however, the relative importance of non-exhaust emissions may increase as electric vehicles become more common-place and if legislation is introduced to reduce exhaust emissions [55]. Vehicle-related pollution levels have high spatial variability and are influenced by many factors, including wind speed, traffic density, and distance from main roads [55]. Many studies have shown that pollution levels decrease rapidly with distance from major roads [53,55–61]. There is now widespread evidence of the adverse health effects associated with exposure to traffic-related air pollution [62–64], including studies finding associations between health risks, such as the increased risk of asthma in children living closer to major roads [65].

Australia lags behind other developed nations in fuel [66] and vehicle emissions standards [67], and there is significant evidence for reduced health costs associated with introducing more stringent fuel [66,68,69] and vehicle emissions [70] standards in Australia. Diesel vehicle emissions are known to be carcinogenic [71] and have been identified as a particular health risk for vulnerable populations, producing toxic ultrafine particles (among other pollutants) that can penetrate deep into lungs and enter the bloodstream [72]. In addition, 'real' (on road) measurements of $NO_X$ and 'hydrocarbon + $NO_X$' emissions by diesel vehicles have been found to be much higher than that reported by laboratory tests [73], increasing the need for stricter controls on the use of diesel fuel. Such controls have already been recommended for NSW for heavy vehicles [74] to limit $NO_X$, and should also be applied to passenger vehicles. The phasing out of diesel vehicles has been shown to make a rapid improvement in air quality, as demonstrated by a detailed modelling study in Europe [75].

An investigation into the health impacts of ethanol-blended petrol [68] concluded that $PM_{2.5}$ emissions from Australian passenger fleet vehicles were reduced when ethanol blends were used compared to pure petrol. Significant health benefits to the Sydney population were estimated from the uptake of ethanol-blended fuels due predominantly to reductions in particulate matter [68], although there is evidence that $O_3$ levels may increase with ethanol use [76]. Improving fuel standards has a proven track record of success in reducing emissions, for example, total emission reductions have been achieved despite significant increases in vehicle miles travelled in the USA [75]. In Australia, modelling of the effects of the Fuel Quality Standards Act 2000, showed that vehicle emissions for 2015 were less than half of the year 2000 levels for CO, $NO_X$, $PM_{2.5}$, $SO_2$, and VOCs [77]. Nevertheless, Australia permits higher levels of sulphur (an important precursor to particle formation) in petrol than other developed nations such as USA, Canada, China, and European countries, and modelling suggests that strengthening fuels standards in 2020 could save health costs of $110 million per year [78]. Recent studies in the USA show a multitude of health benefits following the reduction of sulphur from 30 ppm to 10 ppm (Australia's limit is currently much higher at 150 ppm for low octane fuel) [78]. The revision of Australia's Fuel Quality Standards Regulation commenced on 1 October 2019, and petrol sulphur content will be reduced to 10 ppm by 2027. Lower sulphur levels in vehicle fuels will lead to reduced emissions of $SO_2$, and thus a lesser burden of secondary inorganic aerosols.

Importantly, from 1 January 2020, Australia is obliged to comply with international shipping low sulphur diesel fuel of 0.5% sulphur (or 5000 ppm), which is a substantial reduction on the current permitted levels of 35,000 ppm [79]. Even so, shipping emissions will continue to be a large source of $SO_2$ for Sydney, with local ferry services exempt from the international shipping regulations. Much more stringent controls apply in other locations (e.g., the USA has a 1000 ppm (0.1%) limit for sulphur levels in fuel in control zones, such as harbours [80].

*1.6. Natural Emissions and Their Impact on Air Pollution*

Extremely poor air quality in Sydney, characterised by days in which the 24 h concentration of $PM_{10}$ exceeded the 99% percentile, is often due to bushfire smoke or occasional dust storm events [51]. Many of these fires are the result of prescribed burning (otherwise known as hazard reduction burning) rather than wildfires [51,81]. Of the 52 days rated as having extremely poor air quality from 1994 to 2007, 48 were associated with bushfire smoke. During the same period, 59–90% of episodes where particulate matter concentrations exceeded the national standard were due to fires in Sydney [82]. Bushfire smoke was also the main cause of particulate pollution events in other mainland Australian cities from 1994–2007 [82].

Fire plays an important role in the Australian landscape [83]. Many of the native flora must be burnt to trigger germination of seeds [84]. Prior to European settlement, Indigenous Australians managed the landscape with a mosaic of small fires, and these practices continue, for example, in parts of Northern Australia [85]. In recent decades, planned burning has been adopted in order to reduce the severity and frequency of large uncontrolled fires [86]. Such preventative fires are typically undertaken in spring or autumn, with low winds; however such weather provides little dispersion of the smoke, meaning that nearby communities risk significant acute exposure. This trade-off between fire safety and air quality remains the subject of ongoing debate [87].

An example of a fire event degrading the air quality of Sydney occurred in October 2013. During fires burning to the northwest of Sydney from 17–27 October 2013, air quality stations in Sydney recorded exceedances of the daily mean NEPM for $PM_{10}$ (50 µg m$^{-3}$) and $PM_{2.5}$ (25 µg m$^{-3}$) on seven and nine days, respectively [28]. During the same event, a combination of ground-based measurements, satellite observations and modelled meteorology was used to attribute high particle concentrations in Brisbane (a city approximately 730 km north of Sydney) to smoke from the fires northwest of Sydney [52], transported at high altitudes before intruding to the near-surface. This result demonstrates the potential of regional impacts of large fires in the eastern Australian context.

Air quality in Sydney is occasionally impacted by dust storms originating in the dry regions of inland Australia, most often during spring. A dust storm in October 2005 was due to a combination of high wind speeds and a hot, dry period in western Queensland and NSW preceding the event. $PM_{2.5}$ and $PM_{10}$ concentrations exceeded the NEPM standards in Sydney, Brisbane, Mackay, and Gladstone during the event (encompassing over 1600 km of the eastern coast of Australia) [88], demonstrating the regional impact of dust storms. The most noteworthy recent dust storm impacting Sydney's air quality, known as the 'Red Dawn', occurred during September 2009 and attracted significant international news coverage. During this storm, $PM_{10}$ concentrations greater than 15,000 µg m$^{-3}$ were recorded in the Sydney basin [89], approximately 750 times the background concentration. Daily $PM_{10}$ concentrations above 2000 µg m$^{-3}$ in regional NSW cities were among the highest ever recorded in the literature for Australia. Estimates of the dust mass of this event are in the range of 2.5–3.2 million tonnes [89,90]. In February 2019, when most of eastern Australia was under drought conditions, a major dust storm originating from Central Australia, caused high concentrations of $PM_{10}$ in the Sydney metropolitan area and in many towns and cities in western and northern NSW, including Armidale and Tamworth. The dust was then transported across the Tasman Sea, affecting air quality in the Canterbury region in the South Island of New Zealand [91].

### 1.7. Health Impacts of Air Quality in Sydney

The relationship between air pollution and human health is observed in cities around the globe [25]. Nationally, the estimated deaths per 100,000 population attributed to ambient air pollution is much lower in Australia than in other nations. The age-standardised death rate due to air pollution, (which takes into account differing age structures of the populations), is estimated to be 0.2 deaths per 100,000 persons in Australia (uncertainty interval: $\pm 3 \times 10^{-2}$), compared to 0.3 ($\pm 7 \times 10^{-2}$) in New Zealand, 9 ($\pm 3$) in the United Kingdom, 13 ($\pm 6$) in Germany, and 70 ($\pm 59$) in China [25].

The Australian Institute of Health and Welfare [92] has estimated that in Australia in 2011, 1.6% of all fatalities were attributable to air pollution. Another study, focusing on Sydney, indicated that 2.1% (90% confidence interval: 1.5–2.6%) of deaths were attributable to fine particulate matter and a further 0.8% (95% confidence interval: 0.6–1.1%) due to ozone [9].

Air pollution has health impacts across the life course. It has been associated with adverse birth outcomes, reduced lung function, respiratory and cardiovascular diseases, and more recently, diabetes, autism, dementia, and reduced cognitive function [93–96]. Despite the relatively low levels of $PM_{2.5}$ in Australia, significant associations between air pollution and human health impacts have been observed in Sydney and other parts of Australia [11,22,97–99].

### 1.8. The Influence of Allergenic Pollen

Aerobiological particles (e.g., pollen, fungal spores) also influence the air quality of Sydney [100] and warrant discussion due to their contribution to the triggering of allergic respiratory diseases such as asthma and allergic rhinitis. Representative families of grass and weed pollen have been shown to exhibit positive associations with temperature and wind speed at Bankstown airport in the Sydney basin [101]. One study found eleven synoptic categories for Sydney from October to March [102]. Three of these categories were associated with high concentrations of pollen from Cupressaceae (cypress trees), Oleaceae (olive trees), and Poaceae (grasses). All synoptic categories were associated with strong, dry westerly winds created by a low-pressure system to the south of the Australian continent. Interestingly, high pollen counts were observed over a wide range of temperatures. The most extreme pollen pollution episode in modern times occurred in Melbourne, Australia, in 2016 with nine people killed and 476 people hospitalised during a single thunderstorm asthma event that impacted many people with no previous history of asthmatic problems [103–105]. The risks associated with such events are likely to increase as the impacts of increased atmospheric carbon dioxide levels and climate change are felt in Australia, impacting plant biology, with greater pollen loads and allergenicity predicted [106].

### 1.9. Managing Air Pollutant Exposure

Managing exposure to air pollutants is essential in order to reduce the health impacts of poor air quality. There are three techniques that are currently applied together to achieve this aim. The first is to reduce emissions of air pollutants, which is the most desirable option. This would lead to lower pollutant concentrations in cities (and elsewhere), and thus lower population exposure. The second technique involves moving large emitters of air pollutants away from densely populated areas or ensuring adequate separation distances between sources, such as transport corridors and industrial areas, and sensitive land uses such as residential and commercial areas. These strategies ensure that smaller proportions of vulnerable members of the population are exposed to elevated levels of air pollutants. This requires air quality and health considerations to be integrated into infrastructure, land use and transport planning processes. Finally, individuals can take responsibility for their own exposure and avoid being outside during hazardous periods and other similar behavioural modifications. This underlines the importance of education and of timely access to air quality information, including real-time air quality measurements and accurate air quality forecast information to help susceptible people additionally manage their exposure. Positives and negatives of each technique are summarised in Table 2 below.

**Table 2.** Positives and negatives of some air quality management techniques.

| | | Reducing Emissions | Managing Land Use | Reducing Exposure |
|---|---|---|---|---|
| + | | - climate co-benefit<br>- reduces total environmental impact of the pollutants, not just in the urban air-shed<br>- the most effective way of reducing exposure<br>- stimulates the development of new technology to meet societal energy and material needs while lowering emissions | - does not require new technology<br>- could incorporate green space which may have other co-benefits (including increased visual amenity and reduced urban heat)<br>- reduced conflict between polluting sources and sensitive receptors<br>- cost effective if proactively planned for | - flexible: allows tailoring to each neighbourhood/lifestyle<br>- preventative measure that can help address risks due to unavoidable natural emissions |
| - | | - potentially expensive to implement<br>- delay in pollutant concentration reduction<br>- does not prevent hazardous episodes due to natural causes like dust-storms and bushfires | - expensive to move sources or introduce separation distances after development<br>- no climate co-benefit in terms of reduced greenhouse gas emissions<br>- inequitable as there will always be some population near infrastructure<br>- requires long-term planning | - relies on behaviour modification and public education<br>- no guaranteed exposure reduction (indoor may exceed outdoor pollution)<br>- places responsibility on susceptible individuals to manage exposure rather than on polluting sources<br>- places responsibility on government to provide timely access to air quality information<br>- may be less accessible to low socio-economic groups |

*1.10. Benefits of Air Quality Abatements on Health*

Health impact assessment (HIA) methods have been recently used to assess the impact of changes to emissions on health using different Australian scenarios [9,69]. One such recent study looked at the modelled benefits to human health of improving air quality in Sydney [9]. A hypothetical sustained reduction in 2007 $PM_{2.5}$ concentrations by 10% was estimated to result in 640 (95% CI: 430–850) fewer premature deaths and 3500 additional life-years (95% CI: 2300–4600) in Sydney over 10 years [9]. HIAs targeting changes to emissions from specific sources have also been conducted. For example, an HIA examining the effect of regulating shipping fuel while at port in Sydney reported that ship emissions contributed 1.9% of the population-weighted $PM_{2.5}$ concentration in Sydney in November 2010, resulting in 220 annual years of life lost [69]. Limiting fuel sulphur content to 0.1% while at berth would reduce population exposure to ship emissions by 25% while limiting sulphur content within 300 km of the city would cause $PM_{2.5}$ exposure from shipping emissions to decrease by 56% [69].

Some policies designed to limit greenhouse gas emissions have co-benefits for air quality and therefore, for human health. For example, $PM_{2.5}$, CO, and total hydrocarbon emissions are lowered when a vehicle fleet is fueled with ethanol blends (compared to pure petrol). The potential health cost savings for a 50% uptake of E10 fuel (which blends up to 10% ethanol with unleaded petrol) in Sydney by 2006 was calculated at $16 million per annum and at $17 million per annum for a 100% uptake in E10 by 2011 [68]. Other initiatives have a climate trade-off; for example, the use of low-sulfur fuels leads to lower sulfate aerosols, which have a short-lived radiative cooling effect.

State government-sponsored projects have resulted in the development of marginal abatement cost curves to assess the specific local economic, environmental, and social benefits of various air quality abatement measures across populated areas of NSW [107]. Changes in industrial processes (such as technical changes in the operations of coal-fired power stations) and domestic (such as changes in response to legislation regarding personal vehicle emissions or wood-heater) sources of air pollution were considered [107]. Abatement techniques were prioritised, and specific recommendations were provided for the Greater Metropolitan Region (encompassing Sydney, Newcastle, and Wollongong) and for each city. The high-priority recommendations for Sydney included retrofitting or replacing diesel locomotives, trucks and buses; introducing emissions standards for off road vehicles and for wood heaters; limiting industrial $NO_X$ and $PM_{10}$ emissions and switching to gas electricity generation. For the Greater Metropolitan Region of NSW (GMR), priorities included encouraging more people

to cycle; investigating supply chain sustainability; controlling $NO_X$ emissions from coal fired power stations, and introducing Euro 6 emissions standards for passenger vehicles [107]. Priority actions proposed for investigation by the NSW Government's Clean Air for New South Wales Consultation paper targeted industrial, mining, and coal-fired power station emissions; emissions from on-road and off-road transport; household emissions and specifically wood smoke; and smoke from hazard reduction and open burning [108].

Another potential method to reduce air pollution is to undertake urban greening projects. Green-space has the co-benefits of mitigating the urban heat-island effect, as well as improving the psychological and physical well-being of nearby residents [109]. In one study, measures of particulate pollution (total suspended particles, $PM_{10}$ and $PM_{2.5}$) were negatively correlated with surrounding area greenspace across the Sydney central business district [110], demonstrating the potential positive impact of dense vegetation in urban areas on air quality. However, both the aerodynamic and pollutant removal effects of urban greening must be considered. Modelling studies have found that urban vegetation can limit mixing of pollutants in street canyons and thereby have a detrimental impact on localised urban air quality [111]. To counter this effect, Janhäll [112], informed by a review of the relevant literature, suggests the planting of low vegetation close to emission sources, allowing the improvement of local air quality without limiting mixing in street canyons. Further research is needed on the positive and negative impacts of urban greening on air quality and human health as the existing evidence is mixed [113,114].

It is also important to consider emissions of biogenic VOC species (BVOCs) from the trees themselves. Due to relatively high concentrations of $NO_X$ found in urban areas, ozone production is often limited by the availability of VOCs. Therefore, planting of high BVOC emitting species that make atmospheric VOCs more available could drive up secondary organic aerosol formation and ozone production, especially in hot, dry conditions [19,115,116]. The traits of urban European trees relevant to air quality have been documented [117]; however, very little research has been published for native Australian species [118,119]. This is an important gap as some Australian tree species, including certain species of eucalypts, are high emitters of BVOCs [120]. Further, it is important to consider the potential for increased exposure to pollen when evaluating tree species.

Reducing exposure is the third part of an overall strategy to reduce harm from air pollution. A short-term solution is to remain indoors, especially during high pollution events. This may be applicable during major smoke and dust events, depending on the ventilation of the building [121], but in the wider context, there is a need to reduce ongoing (non-event) exposure at the source, where possible. Further consideration of the ingress of ambient air pollution indoors is needed. Unfortunately, there is limited information on indoor pollution in Australia, (see [122–125] and references therein). Further work, including evaluation of the health impacts of indoor and outdoor air for a range of buildings, residential, commercial, and retail are needed.

*1.11. Climate Change and Air Quality*

Climate change may be a significant driver of changing air quality in Sydney. Stronger predicted near-surface temperature inversions over southeast Australia might intensify poor air quality in the Sydney basin [126] by limiting the atmospheric mixing of locally emitted pollutants. The predicted worsening of air quality in Sydney due to climate change mirrors global predictions, such as the study by Zhang et al. that reported increases in surface maximum 8-h $O_3$ concentration by 22% during a heatwave in the USA. [127]. Similarly, using global chemical models coupled to chemical transport models, Jacob and Winner [128] predicted that increased periods of air mass stagnation and increasing temperatures would lead to increased $O_3$ concentrations in many regions, particularly in polluted urban areas. The authors also highlighted that the increased frequency of wildfires driven by climate change could result in increased particulate concentration separate to the effects modelled using meteorological variables.

Increased atmospheric stability, warmer temperatures, and increased BVOCs may also drive increased secondary aerosol formation in the Sydney basin. Dean and Green [129] performed a review on climate change, air quality, and health impacts in Sydney. They highlighted the likelihood of worsened air quality in a city with a changed climate and identified the need for more focussed research on the response of particulate matter to climate change. Both the total population of Sydney and the proportion of the population that is especially vulnerable to poor air quality episodes are predicted to increase in the coming years (with an aging population and strong population growth predicted in Sydney's west). This will exacerbate existing air quality issues such as those in Western Sydney, where fine particulate matter and $O_3$ concentrations are typically higher than elsewhere in the city. The increased number of people living at the urban–bush interface will increase the potential for people to be impacted by bushfire smoke events (both wildfire and planned burns).

## 2. Key Findings from the Clean Air and Urban Landscape Hub and Its Collaborators

### 2.1. Air Quality Measurement Studies

The CAUL hub formulated a programme of research to provide air quality measurements in New South Wales that could help achieve three main objectives:

1.  To answer questions posed by the public in a series of "road-shows" that were organised when CAUL was established;
2.  To provide novel atmospheric composition measurements that can provide a better understanding of the concentrations of ammonia in the urban atmosphere and the impact of smoke from wildfires, hazard reduction burns, and domestic wood-heaters;
3.  To finalise and publish the atmospheric composition data from a number of previous measurement campaigns so that these could be used for rigorous testing of the performance of different air quality models over New South Wales.

### 2.1.1. Publicly Driven Research

There is increasing public concern about air quality in Sydney. This was reflected in a common question raised at the CAUL "road-show" events, which was some variation of: "are the pollution concentrations at air quality monitoring network sites around Sydney truly representative of what I am exposed to in my everyday settings"? Undeterred by the fact that this is an unanswerable question (due to the huge variability in peoples' daily lives), we set about attempting to provide some insights to this issue via two separate case-studies:

1.  In the first case-study, ambient air quality measurements were made on the roof of a two-story building in the Sydney suburb of Auburn, to evaluate conditions that might represent a typical suburban balcony site. Measurements made at the balcony site were then compared to data from three nearby regulatory air quality monitoring stations [130]. Overall, the air quality at the balcony was similar to that measured at the regulatory sites. Average $O_3$ and $PM_{2.5}$ concentrations were lowest at the Auburn balcony site; nitrogen oxides were within the range measured at the other sites, and carbon monoxide was highest at Auburn. Considering that $O_3$ and $PM_{2.5}$ are the pollutants of most concern in Sydney, we concluded that the existing air quality network provides a satisfactory indication of concentrations of outdoor air quality pollutants at the selected "balcony" site at Auburn.
2.  The second case study examined roadside concentrations of $PM_{2.5}$ during an intensive three-day campaign. $PM_{2.5}$ concentration measurements were made in the vicinity of a major road (known to carry heavy traffic) in the Sydney suburb of Randwick. Observed $PM_{2.5}$ concentrations were compared to regional urban background levels, and the spatial and temporal variations were analysed [131]. This study showed a highly variable spatial distribution of $PM_{2.5}$ along the main road studied. The average $PM_{2.5}$ roadside concentration recorded was 13 $\mu gm^{-3}$, which

was approximately twice the concentration of the nearby regulatory air quality network sites. Those people residing at, (or working for long hours outdoors at), busy roadside locations are, therefore, likely to be at enhanced risk of suffering detrimental health effects associated with air pollution. $PM_{2.5}$ levels were observed to decrease by 30% at a distance of 50 m away from main road intersections, suggesting that pedestrians and cyclists should use side-streets whenever possible. $PM_{2.5}$ concentrations were recorded to be 50% higher in the morning rush hour than the afternoon rush hour at roadside locations, implying that joggers and cyclists can reduce their $PM_{2.5}$ exposure by choosing to exercise in the afternoons rather than the mornings, (although avoiding busy road locations whenever possible is advised).

Our case study in Auburn showed that the New South Wales air quality monitoring network provided a good representation of pollution levels at our chosen "balcony" site. Although this result cannot be generalised to all balcony locations in western Sydney, it does demonstrate the effectiveness of the regional air quality monitoring network in this case. In contrast, average roadside $PM_{2.5}$ concentrations in the sampled areas of Randwick were found to be approximately twice those measured at nearby air quality monitoring stations. We also found very high spatial variability of $PM_{2.5}$ at roadside locations, meaning that roadside air quality cannot simply be evaluated by locating an air quality monitoring station at a single roadside location. Instead, estimates of the average increase of common pollutants at roadside locations (compared to regional background values) are needed to supplement regional air quality monitoring. The heightened concentrations at intersections and near bus-stops should give additional weight to the recommendation of the broad-scale adoption of anti-idling emissions control technologies in on-road motor vehicles, and improvements in road design, such as bus lanes that move the bulk of traffic further from the curb. In future, improved estimates could be made by a network of fixed roadside sensors that operate year-round, but currently, the technology is still developing [132]. From these case studies, we conclude that the existing air quality monitoring network in New South Wales is likely to be fit for purpose, with respect to representing urban background pollutant concentrations, and that outreach programmes should be undertaken to inform the public of simple steps that can be taken to minimise their exposure.

Another suggestion from the CAUL "road-shows" was to carry out a research project looking at urban greening (and mosses in particular) to mitigate air quality impacts. Research in other cities has shown that roadside trees can either decrease or increase local concentrations of air pollutants depending upon the degree to which they hinder the dispersion of pollutants, with hedges being shown to be a particularly good choice of vegetation barrier, and green roofs also an effective air pollution abatement measure [24]. Moss proved to be even more efficient at removing particulate matter from the atmosphere than the nearby native tree species that were tested [133]. This study compared the particulate matter entrapment by roadside moss turfs with that of leaves of a common native tree in the coastal city of Wollongong, NSW, Australia. Plant samples were collected from nine sites on an urban gradient, in three urban classes based on road type: low (quiet roads in peri-urban suburbs), medium (busy suburban roads), and high (freeway-type). Chlorophyll fluorescence, a common measurement of photosynthetic efficiency, was also measured as a proxy for plant stress. By dry weight, moss trapped more than the leaf samples. In addition, greater amounts of total particulate matter were trapped by mosses at the more urbanized sites, implying a positive trend along the urban gradient. The trend in particulate matter trapped by moss was similar to the trend in average ambient $PM_{2.5}$ concentrations measured for two weeks at one site from each urban class, by the deployment of a mobile sensor. The sampled vegetation was also increasingly stressed along the urban gradient (although the exact physical or chemical causes of the stress are unknown): the photosynthetic efficiency of tree leaves declined by 2% from low to high urbanisation, while moss photosynthetic efficiency declined by 40%, indicating a steeper stress gradient for mosses. While the trees appeared to be less affected, both plant types appeared to respond to urbanisation by increasing wax deposition [134].

A companion study investigated the comparative ability of four different indigenous tree species in NSW to remove particulate matter from the atmosphere. This study showed that evergreen trees

absorb particulate matter into their leaves, whilst in deciduous trees, the particulate matter deposits onto the leaf surface and can get washed off. This means that deciduous trees' ability to scavenge particles gets renewed after rainfall events [134].

2.1.2. Novel Atmospheric Composition Measurements in Sydney

In conjunction with the suburban balcony case-study described above, novel atmospheric composition measurements were made along an integrated open-path of nearly 400 m between a Fourier transform infrared spectrometer and an array of mirrors, across the centre of Auburn. The spectrometer operated for approximately nine months between October 2016 and September 2017, (with a break between March and May 2017) and made routine measurements of carbon dioxide ($CO_2$), CO, methane ($CH_4$), and ammonia ($NH_3$). Concentrations of methanol ($CH_3OH$), acetylene ($C_2H_2$), ethylene ($C_2H_4$), and formaldehyde ($CH_2O$) were also measured during episodes of enhanced pollution.

This novel technique allowed for average concentrations of $NH_3$ to be measured across the open-path, with observed concentrations varying from 1 to 20 ppb, and showed the importance of $NH_3$ from traffic in particulate formation potential in Sydney [135].

By examining the ratios of the observed concentrations of $NH_3$ to CO, this study established that:

- $NH_3$:CO ratios were strongly correlated with traffic volumes on nearby roads, implying that the main source of $NH_3$ at the site is from traffic exhaust fumes, via the operation of catalytic converters. (The $NH_3$:CO ratio will decrease if the emissions are not fresh due to the shorter atmospheric lifetime of $NH_3$ compared to CO.)
- The current emissions inventory for New South Wales (the GMR2008 [20]) underestimates gas-phase $NH_3$ vehicle emissions (when compared to CO) by approximately 40%.
- The urban concentrations observed in this study imply that $NH_3$ is the limiting reagent for production of $NH_4NO_3$ aerosol, but for $(NH_4)_2SO_4$, $SO_2$ is the limiting reagent [135]. This finding provides further evidence to support changing the legislation to reduce the maximum permitted sulfur levels in shipping fuels and vehicle petroleum.

The novel, open-path spectrometer measurements also allowed for the impact of smoke from hazard reduction burns and from domestic wood-heaters to be explored, to answer the following questions:

1. Are there significant differences in the chemical composition of smoke from domestic wood-heaters and smoke from hazard reduction burns?
2. During the "balcony" case-study, which of these sources of wood-smoke caused the greatest exposure to pollutants of concern (e.g., $PM_{2.5}$) in Auburn?

Analysis of the smoke pollution events determined that (for the measured components) the chemical composition of smoke was very similar, whether the smoke originated from domestic wood-heaters or from hazard reduction burns. During the study period in Auburn, hazard reduction burns were a greater immediate acute threat to public health (because peak concentrations of particulate matter were highest during these events). However, domestic wood-heater events produced greater cumulative exposure during the campaign, due to the greater duration of enhanced pollution from this source, so, also presenting a public health threat. Whilst both of these pollution sources vary from year to year, this study highlighted the significance of pollution from domestic wood-heaters in Sydney as an issue of importance both for the public and for policy-makers.

Outside of the major capital cities, Australia is a large and sparsely populated nation. Most of the air quality monitoring activities are undertaken in the main cities; however, remote-sensing technologies make it possible to expand some of this coverage to the whole population. Satellite-derived estimates of the total-column aerosol optical thickness (AOT) and tropospheric $NO_2$ column density have been shown to be sensitive to concentrations in the boundary layer, and for both aerosols and $NO_2$, concentrations tend to be highest in the lowest layers of the atmosphere [136]. This has been successfully combined within land-use regression models to predict monthly or annually-averaged $NO_2$ or $PM_{2.5}$

concentrations at the earth's surface [137,138]. Such methods have the advantage that they are not limited by state or local government boundaries and offer relative consistency in their method.

### 2.1.3. Findings from Previous Measurement Campaigns in New South Wales

An important aspect of air quality management within a city is the accuracy and reliability of the operational air quality model for the region. The establishment of the CAUL hub provided the opportunity to undertake a major air quality modelling comparison that could benchmark the available air quality models against each other and against previously established standards for performance (see Section 2.2). However, in order to elucidate different aspects of why one model may outperform another, it is important to have very detailed atmospheric composition data that includes a range of species that are not routinely measured as part of the regulatory air quality monitoring network. The Australian atmospheric chemistry community had undertaken three relevant measurement campaigns to gather detailed atmospheric composition data, but these data had not been through their final quality assurance procedures that were required before publication. In the early stages of CAUL, efforts were concentrated on finalising these datasets from the two Sydney Particle Study campaigns (SPS1 and SPS2) [32–34] and from the Measurements of Urban, Marine, and Biogenic Air (MUMBA) campaign [6,139,140] from the southern city of Wollongong. These three campaigns were used as the basis of the modelling comparison described in Section 2.2. In addition, work has been undertaken to finalise the data gathered during two campaigns in the industrialised Hunter Valley of New South Wales [141].

The SPS1 and SPS2 campaign findings are published elsewhere [33]; here, we summarise the findings from the other campaigns that were reported in this special issue.

The MUMBA campaign ran for eight weeks from mid-December 2012 to mid-February 2013, providing a rich dataset of atmospheric composition at the marine/urban/forest interface. An episode of clean marine air enabled the measurement of background concentrations of key species (including a number of VOCs) at these latitudes [139]. MUMBA also provided interesting observations of isoprene from nearby vegetation and other biogenic VOCs, which predominantly originated from the forested escarpment that surrounds much of Wollongong. These natural biogenic emissions play an important role in the control of air quality in Australian cities, due to the remoteness from other polluting sources and the relatively low local anthropogenic emissions [139].

Wollongong is well known in New South Wales as an industrial city; nevertheless, traffic emissions were shown to be the main driver of $NO_X$ concentrations at the MUMBA site, and the air-shed was VOC-limited [6]. The daily average mass concentration of fine particles ($PM_{2.5}$) was low (6.1 $\mu$g m$^{-3}$) during the MUMBA campaign [142]. The particle number concentration was dominated by ultrafine particles (particles with a diameter between 3 and 100 nm), with an average and median value of $7.0 \times 10^3$ cm$^{-3}$ and $5.2 \times 10^3$ cm$^{-3}$, respectively. Eight particle formation and growth events were identified from the particle number size distribution range from 14 nm to 600 nm dataset [142]. Particle formation and growth events occurred in air masses that travelled from the ocean and passed through populated areas, including Sydney. Anthropogenic sulphate, the photochemical age of air masses and relative humidity potentially played a role in the particle formation and growth events. Sources of particles identified included traffic emissions, industrial activities, and the marine atmosphere [142].

There were two days of extreme heat during the MUMBA campaign (with temperatures exceeding 40 °C). This provided a good test case to model the impact of extreme temperatures on $O_3$ formation in Sydney [143]. Ratios of biogenic VOCs measured at the MUMBA site on these hot days were different from those measured on other days of the campaign, but further measurements are needed to understand to what degree this resulted from the different vegetation types being sampled and how much was caused by the extreme heat [139].

Further north of Sydney is the Hunter Valley, where domestic housing is located close to major industrial sources, and communities are concerned about the impact of coal mining and associated industrial emissions on their health. Such public concerns resulted in two studies in the Hunter Valley

looking at the chemical composition of particulate matter [141,144,145]. Two sampling periods (2012 and 2014) at six sites in the Hunter Valley and across two size fractions ($PM_{2.5}$ and $PM_{2.5-10}$) were input to a receptor model, to determine the source of particulate matter influencing particle composition at the sites. Fourteen factors were found to contribute to particle mass [141]. Of these, three source profiles common to all sites, size fractions, and sampling periods were Sea Salt, Industry-Aged Sea Salt, and Soil. Four source profiles were common across all sites for $PM_{2.5}$, including Secondary Sulphate, Secondary Nitrate, Mixed Industry/Vehicles, and Woodsmoke. One source profile (Other Biomass Smoke) was only identified in $PM_{2.5}$ at the two sites furthest from the coast, and two source profiles (Mixed Industry/Shipping and Vehicles) were only identified in $PM_{2.5}$ at the four sites closest to the coast [141].

The contribution of the Soil Factor to $PM_{2.5}$ mass is consistently about 10% of the total mass across the sites, while Sea Salt decreased with distance from the coast and Smoke increased with distance from the coast, (but together these two classes make-up about 25–50% of the mass of $PM_{2.5}$ at all sites) [141]. The largest contribution to $PM_{2.5}$ was from industrial sources (primary and secondary) at all except the most inland site, where Woodsmoke and industry sources made an equal contribution of 40% with most of the industry component from secondary processes (80%). At most sites, primary emissions accounted for approximately 30%, and secondary reactions accounted for approximately 70% of the industry source [141]. Studies that identify the major sources of atmospheric fine particulate matter are extremely useful in helping to prioritise efforts to reduce atmospheric concentrations by emissions control measures.

## 2.2. CAUL Air Quality Modelling Comparison and Modelling Studies

Air quality modelling is an important aspect of the management of atmospheric pollution in any community. It allows for public warnings to be issued when air quality is predicted to be poor, as well as providing insights to the causes of different pollution events and analysis of the impact of future emission scenarios. A major undertaking within CAUL was the first comparison of hourly air quality models over Sydney, using a suite of six air quality modelling systems, over the time periods of the SPS1, SPS2, and MUMBA campaigns. The comparison resulted in improvements to the implementation of models over Sydney and demonstrated that air quality modelling over the greater metropolitan regions of New South Wales can meet international standards of performance [146,147].

Such modelling comparisons help cross-validate the models, test their skill at reproducing observed atmospheric composition, and identify any flaws or problems in the way that the models are set up or run [53,147]. At the end of the comparison exercise, the validated models may be used to undertake a number of different studies with added confidence in the modelled output [143,148,149].

The air quality model comparison used two separate meteorological models (CCAM and WRF) with a total of seven different configurations and was conducted over consistent geographical domains, grid resolutions, and time periods [147]. The modelling domains were nested so that the outer grid covered the whole of Australia at 80 km resolution, whilst the innermost (of four) grids covered the Sydney basin at 3-km resolution. Comparison of the meteorology within the models identified systematic overestimates of wind speeds that were more pronounced overnight, which is a common weather model bias [147]. The temperatures were well simulated, with the largest biases also seen overnight. The models tended to have a drier lower atmosphere than observed, implying that better representations of soil moisture and surface moisture fluxes would improve the subsequent air quality simulations [147]. The local-scale meteorological features, such as the sea breeze, which is a critical feature driving ozone formation in the Sydney Basin was reasonably well-represented in the simulations. Overall, the biases between simulations and observations were generally within the recommended benchmark values with the exception of extreme (both high and low) events, when the biases tended to be larger [147].

The main driver of the interaction of meteorology and air quality is the degree of atmospheric mixing that acts to dilute ground-level emissions of primary pollutants. Chambers et al., 2019 [5], show

the usefulness of radon as a tool for inferring atmospheric boundary layer heights to better constrain the atmospheric mixing within air quality models. In this study, the modelling comparison results were evaluated within different atmospheric "class-types" defined over 24-h periods using a $^{222}$Rn-based stability classification technique [5]. Calculating hourly distributions of observed and simulated quantities within each class-type helped: (i) bridge the scale gap between simulations and observations, (ii) separately represent the variability associated with spatial and temporal source heterogeneity rather than it adding to bias values, and (iii) provide an objective way to group results over whole diurnal cycles that separates uncontrollable sources of uncertainty (synoptic non-stationarity, rainfall, mesoscale motions, extreme stability, etc.) from parameterisation problems, or between-model differences [5]. Meteorological model skill varied across the diurnal cycle for all seven models, with an additional dependence on the atmospheric mixing class that varied between models. Model skill regarding air quality varied strongly as a function of mixing class and was typically worst when public exposure would have been the highest (during episodes of poor air quality). This has important implications for using contemporary models to assess potential health risks in new and rapidly evolving urban regions [5].

The CAUL hub was particularly interested in exploring Indigenous knowledge and perspectives. The annual cycles in meteorological variables in Sydney were used to identify a set of quasi-seasons using a combination of Indigenous knowledge, statistics, and historical data from the Bureau of Meteorology in Australia [150]. This approach was particularly successful in identifying the coldest time of year, when atmospheric mixing is at its lowest, and there are peak concentrations of $PM_{2.5}$, CO, and $NO_X$ in Sydney. The methodology used could easily be applied in other parts of the world [150].

Improvements to air quality forecasts may be gained by increasing the complexity of models (such as coupling to ocean models), although at the cost of greater computing resources [151,152]. The relative performance of the Weather Research and Forecasting model with chemistry (WRF/Chem), with and without coupling to the Regional Ocean Model System (ROMS) (WRF/Chem-ROMS), was shown in two paired papers [151,152]. WRF/Chem-ROMS generally performs well at 3-, 9-, and 27-km resolutions for sea-surface temperature and boundary layer meteorology, despite larger under-predictions for total precipitation due to the limitations of the cloud microphysics scheme or cumulus parameterisation [152]. The model also performs well for surface $O_3$, under-predicts $PM_{2.5}$ and $PM_{10}$ during SPS1 and MUMBA and over-predicts $PM_{2.5}$ and under-predicts $PM_{10}$ during SPS2. These biases are attributed to inaccurate meteorology, precursor emissions, insufficient $SO_2$ conversion to sulphate, inadequate dispersion at finer grid resolutions, and under-prediction in secondary organic aerosol [152]. The use of finer grid resolutions (3- or 9-km) can generally improve the performance for most variables.

In the companion paper [151], the performance of WRF/Chem and WRF/Chem-ROMS are compared for their applications in Australia. The explicit air-sea interactions in WRF/Chem-ROMS led to substantial improvements in simulated sea-surface temperature, latent heat fluxes, and sensible heat fluxes over the ocean during all three field campaigns, which led to better performance of WRF/Chem-ROMS in boundary layer meteorology [152]. The percentage differences in simulated surface concentrations between the two models were mostly in the range of ±10% for CO, OH, and $O_3$, ±25% for $CH_2O$, ±30% for $NO_2$, ±35% for hydrogen peroxide, ±50% for $SO_2$, ±60% for isoprene and terpenes, ±15% for $PM_{2.5}$, and ±12% for $PM_{10}$ [151]. The satellite-constrained chemical boundary conditions reduced the model biases of surface CO, NO, and $O_3$ predictions at 3-km for all field campaigns, surface $PM_{2.5}$ predictions at 3-km for SPS1 and MUMBA, and surface $PM_{10}$ predictions at all grid resolutions for all field campaigns. The chemical boundary conditions were shown to be more important in the relatively clean Southern Hemisphere, than in the more polluted Northern Hemisphere [151].

The continued monitoring of air quality in Sydney has shown that the city experiences exceedances for only two of the regulated pollutants, $O_3$ and $PM_{2.5}$. For this reason, the final paper from the comparison exercise presents the overall performance of the six air quality modelling systems in predicting $O_3$ and $PM_{2.5}$, during the SPS1, SPS2, and MUMBA campaigns [153]. Model performance

for $O_3$ was evaluated against measurements at 16 air quality monitoring stations. Performance for domain-wide hourly $O_3$ was good, with the models generally meeting benchmark criteria for normalised mean bias (<15%) and correlation (>0.5). The models also reproduced the observed $O_3$ production regime (based on the $O_3/NO_X$ indicator) at 80% or more of the air quality monitoring sites. When the model output is paired with the observations, all models tend to overestimate the lowest observed hourly $O_3$ values and underestimate the highest observed hourly $O_3$ values; as has been observed in other comparison exercises (e.g., [154]). The probability of the models predicting daily maximum $O_3$ values above 60 ppb at specific sites was generally low (0–67%). This probability increased to 25–80% when testing the models for daily maximum $O_3$ values above 60 ppb in a specific region (e.g., Sydney East, Sydney North-West). Relaxing the test further to domain-wide detection of events only marginally improved the probability of detection (28–93%) but greatly reduced the number of false alarms, with the false alarm ratio decreasing each time the test was relaxed (False alarm ratio, domain-wide: 10–40%; region: 32–73%; site: 40–100%).

Performance for $PM_{2.5}$ was assessed using measurements at five air quality monitoring stations during the summer campaigns (SPS1 and MUMBA) and four stations during the autumn campaign (SPS2). Domain-wide model performance for daily $PM_{2.5}$ (24-h averages) was variable. Most models underestimated $PM_{2.5}$ concentrations during the summer campaigns and overestimated them in autumn (SPS2). The benchmark criteria for normalised mean bias (<30%) was met by only one model for SPS2 and MUMBA. Most models met the criteria for SPS1. All models met the criteria for correlation (>0.4) during SPS2, and most did during the summer campaigns. The evaluation of the performance of the models for $PM_{2.5}$ was hindered by the few monitoring sites reporting $PM_{2.5}$ at the time of the campaigns.

As with many other parts of the world, key challenges remain for air quality modelling, including access to accurate emissions inventories and meteorology for the modelled region [6,147]. Australia lacks a consistent national emissions inventory. Only sporadically updated regional inventories of varying resolution, composition, and methodology are produced around some of the cities. As a contribution towards improving emissions inventories, an uncertainty analysis has been made of emissions estimates in the NSW Environment Protection Authority's Air Emissions Inventory for 2008 for the Greater Metropolitan Region [20,155].

After the completion of the comparison study, when the models have been optimized, and their performance has been validated against observations and the ensemble of other models, it is then possible to benchmark the models against international performance standards and apply the models to address particular issues of interest.

Chang et al., 2018, assessed the ability of one of the models from the comparison study (the regional air quality model, the coupled Conformal Cubic Atmospheric Model and Chemical Transport Model (CCAM-CTM)) for the NSW Greater Metropolitan Region to predict concentrations of $PM_{2.5}$, $O_3$, and $NO_2$ by evaluation against air quality data from the NSW DPIE air quality monitoring network [146]. Overall, CCAM-CTM performance was shown to be comparable to that of other regional air-shed models reported in the literature [146]. Generally, the model slightly over-predicted $PM_{2.5}$ concentrations but under-predicted peak values on the most polluted days [146]. The speciation of $PM_{2.5}$ was generally well captured (with a factor of 2) but with some underestimation of the contribution of sea-salt, ammonia, and elemental carbon. $NO_2$ and peak $O_3$ values were also slightly under-predicted. The study also identified possible mechanisms for future improvements in the model, including better characterization of highly variable emissions sources, such as domestic wood-heaters, traffic, and industrial emissions [146].

A study utilising the '45 and Up' cohort [156] estimated the exposure to $PM_{2.5}$ and $NO_X$ for participants who lived in Western Sydney at the baseline of that study [96]. Exposure assessment for $NO_2$ was based on a satellite-based land use regression model, and $PM_{2.5}$ exposure was based upon a Chemical Transport Model (CTM) [99]. The associations between exposure to $PM_{2.5}$ and $NO_2$ at baseline, with hospitalisation for all respiratory diseases over a seven-year follow-up, was assessed.

The median annual concentration of $PM_{2.5}$ was slightly lower for Western Sydney residents compared with the rest of Sydney (4.1 µg m$^{-3}$ vs. 4.6 µg m$^{-3}$); the maximum $PM_{2.5}$ concentrations were higher for residents in Western Sydney compared with other areas in Sydney (13.8 µg m$^{-3}$ vs. 8.11 µg m$^{-3}$) [99]. Median annual concentration of $NO_2$ was lower in Western Sydney compared with other areas in Sydney. Similar to the results for the whole of Sydney [99], no associations between exposure to air pollutants and hospitalisation for all respiratory diseases in Western Sydney were found [99].

$O_3$ formation can be both $NO_X$ or VOC-limited, with the prevalence of a $NO_X$-limited regime over Sydney confirmed during days of elevated ozone concentrations during a 2013 heatwave using the WRF-Chem model (as used in the modelling comparison exercise described above) [143]. However, they also highlighted the importance of biogenic VOC emissions (from Eucalyptus trees, which are prevalent in the forested regions surrounding Sydney), by showing that when all biogenic VOC emissions were removed from the model, no $O_3$ events occurred [143]. This study also highlights that problems with $O_3$ pollution in Sydney are likely to be exacerbated in future years by a warming climate, with higher temperatures increasing emissions of biogenic precursors and speeding up the chemical production of $O_3$ in approximately equal measures. Similarly, being a $NO_X$ limited regime, the benefits of all policy actions to reduce $NO_X$ will simultaneously also reduce $O_3$ (i.e., traffic controls and the mitigation of industrial emissions).

Two further papers use the CCAM-CTM model to identify the major source contributions to $O_3$ and $PM_{2.5}$ in Sydney [148,149]. The most significant contributions to ozone in NSW come from biogenic VOC emissions, which dominate over anthropogenic emissions [149]. The relative importance of different emissions varies between geographic regions of NSW, depending on the ozone formation potential of the region and whether it is $NO_X$ or VOC limited [149]. Commercial and domestic sources are the largest anthropogenic contributor to ozone concentrations because they combine high VOC and low $NO_X$ emissions (except for domestic wood-smoke—which is also a high $NO_X$ emitter) [149]. Emissions controls on power stations will reduce ozone concentrations in North West Sydney, the Lower Hunter, and Illawarra regions of NSW, whilst traffic emissions control will be the most effective policy to reduce ozone in the South West of Sydney, which is most prone to smog and ozone exceedances [149].

To assess the impact on residents, the modelled annual averaged $PM_{2.5}$ concentrations from CCAM-CTM were weighted by the population density (using the 1 km resolution gridded population data from the Australian Bureau of Statistics), [148]. It was found that 60% of the $PM_{2.5}$ burden in the NSW Greater Metropolitan Region originates from natural sources (biogenic emissions, sea salt, and wind-blown dust) and 40% from anthropogenic sources. Of the anthropogenic sources, the most significant contributions to overall population-weighted $PM_{2.5}$ in the NSW Greater Metropolitan Region come from wood-heaters (31%), industry (26%), on-road motor vehicles (19%), power stations (17%), and non-road diesel and marine (6%) [148].

These modelling studies provide evidence for policy-makers of the most important source contributors of two of the pollutants of most concern in NSW ($O_3$ and $PM_{2.5}$). Such studies provide a sound scientific basis for prioritising air quality management interventions to optimise improved public health outcomes.

## 3. Implications for Policy Makers

### 3.1. Policy Options to Minimise Poor Air Quality Episodes from Smoke Pollution

During the CAUL measurement campaign at Auburn, all the major pollution events were associated with fires [130,157], reinforcing previous findings that fires (both wild and prescribed) are responsible for most of the worst air quality events in Sydney [51,82]. The new NEPM excludes hazard reduction burns (incorrectly assigning them to natural episodes) [7]. This discourages the consideration of the health impacts of hazard reduction burns during the planning of such events, to the detriment of the health of the population of Sydney. In order to minimise avoidable severe pollution events, the NEPM guidelines need to be revised so that only exceedances caused by wildfire pollution can be

excluded from the reported exceedance count. Additionally, the use of the latest modelling tools to predict the impact of the prescribed burns on air quality (e.g., [158]) should be mandated. Further research support should be given to improving these tools and communicating these risks to the public. The best possible predictions of smoke impacts should be available before decisions are made on whether or not to ignite a hazard reduction burn. Research support that enables a proper analysis of the pros and cons of hazard reduction burning strategies should continue (like the current DPIE bushfire risk management research hub). The DPIE bushfire risk management research hub brings together researchers, fire agencies and public land managers in a collaborative research effort to improve fire management strategies, and reduce the risk fires pose to people, property and the environment.

Pollution from wood smoke contains a number of additional chemicals (such as formaldehyde and ammonia) that are known human toxins [157]. These will exacerbate the health impacts of exposure to smoke from the legislated pollutants. The chemical composition of smoke is very similar, regardless of whether the source is bushfires or domestic wood-heaters [157]. Wood-heaters are a significant contribution to human sources of $PM_{2.5}$, as discussed earlier [130,148], and measurements made during the CAUL Auburn campaign, established that the overall exposure to heightened pollution levels from smoke from domestic wood-heaters in Auburn was greater than from bushfires, despite five major bushfire pollution events during spring 2017 [157].

We recommend that legislation be considered that works towards eliminating the use of wood-heaters in urban areas (and reducing their use in regional areas prone to meteorological conditions that trap the pollution near the surface). As this legislation is phased in, interim measures could be implemented that further reduce the permitted emissions from the current guidelines/standard of 1.5 g of particles per kg of fuel burned [159].

*3.2. Policy Options to Reduce Air Pollution from Traffic*

As discussed above for wood smoke pollution, preventing exceedances of air quality standards is only part of the challenge, since $PM_{2.5}$ has been shown to cause adverse health impacts, and has no known safe limit [160]. Problems associated with chronic exposure to air pollutants, mean that the best overall outcomes for population health will be via a reduction in emissions from all the major pollutant sources.

Research in this special issue shows that more than half the $PM_{2.5}$ burden in Sydney is from natural sources [148]. Amongst the anthropogenic contribution, emissions from on-road petrol and diesel vehicles account 19% of the population-weighted annual average $PM_{2.5}$ concentrations in the NSW Greater Metropolitan Region [148]. In addition, the CAUL roadside campaign in Randwick [131] confirmed the findings of many international studies that have reported significant increases in $PM_{2.5}$ pollution near busy roads [78,123,161,162]. Many people spend time near traffic hotspots and main roads during their commute or other parts of their day-to-day lives, and so are exposed to heightened levels of pollution above that observed by the regulatory air quality monitoring sites. Furthermore, infill urban development in Sydney is disproportionately occurring along main road sites, so that population exposure to traffic-related air pollution is likely to increase over the next decade. There is also a tendency for this to impact the more vulnerable in our society with those of lower socio-economic status being more likely to live by busy roads and more likely to commute by bus (with bus-stops having been identified as typical pollution hotspots [131,163,164]). Thus, a reduction of emissions from road vehicles is an important issue for public health and social equity.

Regulation of motor vehicles and fuels has reduced air emissions and air pollution concentrations in recent decades despite the growth in traffic activity [108]. However, recent air quality trends and emission projections indicate that air pollution from vehicles will plateau and then increase without further action to protect air quality [165]. We recommend that policies be implemented to reduce traffic-related pollution. Strategies that should be considered include:

1.  Prioritising policies that encourage active transport, such as better pedestrian and separated cycle paths [166,167]. Providing better public transport and considering fiscal policies, such

as introducing congestion taxes and tax deductions for public transport, whilst removing incentives/tax breaks for company cars.

2. Legislation and measures to encourage a more rapid move to low and zero tail-pipe emission vehicles. Australia could follow the lead of nations that have incentivised the uptake of electric vehicles, with some nations also declaring timelines for bans on the sale of new internal combustion engine passenger vehicles, with the aim of improving air quality and reducing greenhouse gas emissions; for example the United Kingdom [168] and France [169]. This will require supporting actions to ensure that the required infrastructure is in place, such as mandating the phasing in of recharge stations at premises licenced to sell petrol and diesel; incentivising the provision of charging infrastructure by the private sector and revising planning instruments and building construction requirements to accommodate infrastructure. The NSW Electric and Hybrid Vehicle Plan released in 2018 [170] commits the state government to co-invest in charging infrastructure on major regional routes and to provide support for charging through strategic land use planning and guides. This plan provides the platform to advocate for further actions to accelerate the move to low and zero tail-pipe emission vehicles. The use of renewable energy to power electric vehicle infrastructure should be maximised to reduce the reliance of fossil fuel power generation. Policies should be put in place to develop a sustainable framework for charging of electric vehicles so as to reduce overall emissions and avoid adverse impacts on the electricity grid [171,172]. Biogenic VOC emissions have been shown to undergo chemical reactions with anthropogenic $NO_X$ in the atmosphere leading to a major source of $PM_{2.5}$ and $O_3$ [148]; however, these natural emissions form particulate matter and $O_3$ after reacting with $NO_X$. Vehicles contribute over 80% of $NO_X$ emissions in the Sydney region [13]; hence, the move to zero tail-pipe emission vehicles may also reduce the apparent contribution to $PM_{2.5}$ and $O_3$ from natural sources of VOCs, in addition to removing the more direct emissions amounting to >20% of $PM_{2.5}$ [13]. Policies to promote the use of electric vehicles should be co-designed with policies to improve the public transport system and encourage its use.

3. During the transition to zero tail-pipe emission vehicles, reduced pollution can be achieved by introducing a further tightening of fuel efficiency, fuel quality, and emission standards, introducing anti-idling control technologies, and by phasing out diesel vehicles [66]. Consideration should be given to addressing non-exhaust emissions, such as tire and brake wear particles and raised dust, which will eventually become more significant as the move to electric vehicles nears completion [165].

4. Set an example by limiting government vehicles and public transport to non-fossil-fuel use. As noted above, vehicle exhaust emissions contribute significantly to criteria air pollutant emissions in the Greater Metropolitan Region. The move away from this engine type, towards low/no emission electric or fuel cell vehicles will provide air quality benefits. NSW has a 10 per cent target for new NSW Government general purpose passenger fleet cars purchased or leased by state agencies to be electric or hybrid vehicles by 2020/21 [170]. This target should be increased for future years and consideration given to the transition of public transport. The large-scale use of electric buses has been successful in cities such as Hefei (>600 buses) and Shenzhen (>1000 buses), China [77]. All levels of governments can contribute to this effort via schemes to promote clean transport and energy generation and by leading by example (e.g., by using electric trains, buses, and motor-vehicles and by installing solar power). This 'early adopter' policy would help bring forward the installation of new infrastructure (e.g., charging stations) required for lower emissions vehicles and would result in a greater number of low/no emission vehicle models being available. This move needs to be coupled with the provision of electricity from renewable sources.

5. Limit motor vehicle engine idling. This has co-benefits for reduced fuel costs and $CO_2$ emissions. It is also particularly effective for air quality since much idling occurs at exposure hotspots such as intersections and car-parks.

*3.3. Policy Options to Reduce Other Major Pollution Sources*

1.  Implement policies to further improve energy efficiency and accelerate the transition to clean energy; so, mitigating air pollution and greenhouse gas emissions from traditional coal-fired power generation. This makes sense for economic reasons also, since the cost of renewable energy is falling rapidly.
2.  Shipping—from January 2020 fuels with less than 0.5% sulphur will be mandated by international shipping laws, but local ferry services like Sydney Ferries are exempt. Modelling has shown large human health benefits could be gained from stricter emissions controls on shipping in Sydney [69]. Thus, a move to overcome the State versus Federal barriers to enforcing ship emissions should be prioritised. Switching to Liquefied Natural Gas or electric, such as is being adopted for ferries in Norway and New Zealand, would see a significant reduction in both greenhouse gases and criteria pollutants [64,65]. Additionally, in-harbour emission for on-board power generation can potentially be mitigated through the provision of electrical mains shore-power for ships when docked.
3.  Control off-road vehicle emissions, which are growing in contribution due to the absence of non-road diesel emission standards. Also address VOC emissions from the commercial and domestic sector, which are emerging as an increasingly important source of ozone and secondary organic aerosol precursors [173].

*3.4. Urban Design to Reduce Exposure*

Urban greening can help reduce fine particulate matter pollution by intercepting and removing aerosols from the atmosphere. More research is needed in this area because of the complex nature of atmospheric composition and sources (e.g., biogenic emissions are precursors to both particulate matter and ozone), and in certain circumstances, trees and bushes can cause a street canyon effect, reducing natural ventilation and trapping pollution closer to the ground where people breathe. Nevertheless, Sydney is surrounded by great swathes of heavily forested regions, and, thus, the extra contribution to biogenic precursors from urban greening projects into the air-shed is unlikely to be significant (although further research is needed).

1.  Look for the best native species for intercepting particulate pollution [134] and with the lowest emissions of allergic pollens and biogenic VOC species most prone to contribute to fine particulate matter and $O_3$ formation.
2.  Consider further trials of moss beds and moss walls, which have been shown in our preliminary study to be efficient at removing particulate matter from the atmosphere [133].
3.  As urban density increases, it will be important to ensure that there are sufficient green spaces both for air quality and for other aspects of livability, including for the mitigation of urban heat [174]. Given the importance of $PM_{2.5}$ in overall air quality, and the evidence of decreasing concentrations of $PM_{2.5}$ with height above ground (supported by the findings of the CAUL Auburn campaign reported in this special issue [130]), there is evidence to support urban planning that encourages high-rise buildings set in ample green-space. This would also be beneficial for walkability and access to public transport [166,167,175].
4.  Use planning permissions to avoid building pre-schools, child-care centres, schools, hospitals and aged care homes near major roads or traffic hotspots or in valleys prone to conditions that trap pollution near the ground since proximity to major roads has been shown to increase exposure to air pollutants [78,161,162,176–178].
5.  Similarly, the location of new polluting industries should consider prevailing wind direction and the relative locations of populated areas (as is done through the NSW EPA approval methods).
6.  Indoor air considerations should not be forgotten in urban design. Design and maintenance at schools should prioritise the transition away from combustion or unflued gas heating, which contributes to poor indoor air quality and is of particular concern in school classrooms.

In addition, classrooms which are inadequately ventilated increase exposure to indoor air pollutants [179]. Air pollution in classrooms can impact the health [180], attendance, and even academic performance of students [181]. Airlocks between attached garages and the living zones of residential buildings should be mandated to prevent direct ingress of vehicle exhaust [182].

7.  Application of modern building codes will provide insulation to reduce cooling/heating costs. Schemes to encourage retro-fitting older properties for similar gains should be encouraged; however, adequate ventilation should also be considered to minimise the build-up of indoor air pollutants and mildew.

8.  Special consideration should be given to ensuring sufficient urban greenery and the planting of trees/bushes as a mitigation measure for fine particulates in the development associated with the new Western Sydney Airport at Badgerys Creek. This is especially important in the west of Sydney due to meteorological conditions that can trap pollution near the surface and exacerbate poor air quality in the west of the city.

### 3.5. Air Quality Monitoring, Modelling, and Public Alerts

Continued monitoring of air quality is crucial for understanding changes in air quality. The Sydney basin has a large network of air quality monitoring stations operated by the Department of Planning, Industry, and Environment (DPIE). The CAUL Auburn study demonstrated that the regional monitoring stations provided a good representation of air quality at a case study site chosen to be typical of a suburban balcony in western Sydney [130]. However, air quality is highly spatially variable, evidenced in the gradient of fine particulate matter from busy road to regional background in 300 m [59]. From the CAUL Randwick campaign, Wadlow et al., [131] concluded that the PM$_{2.5}$ concentrations were very variable near major roads and that hotspots existed by busy intersection and bus-stops. They concluded that the addition of one or two roadside air quality stations to the DPIE network was likely to be less informative than supplementing the existing network with a dense array of sensors around one or two areas to illustrate likely hotspots (and then disseminate this information via a public education program) [131]. Data from roadside air quality stations and sensor networks in hotspots can also be used to 'train' air quality models and thus, provide supplementary information. Careful and strategic planning should be conducted in order to determine the placement and capability requirements of different types of sensors. Roadside or low-cost monitoring should be accompanied by public education; otherwise, the data is likely to be misunderstood and may hinder rather than help the situation.

The DPIE also runs an operational air quality model issuing daily forecasts for the next day and undertakes research to make continued improvements to its performance [146]. Continued efforts to improving air quality modelling should be made as this is of paramount importance to issuing accurate and reliable alerts to vulnerable members of the population. It is important that poor air quality events are anticipated so that vulnerable people can take precautions to mitigate the effects (e.g., seek shelter in an air-conditioned building and take medications as necessary). False alerts must be minimised so that people respond appropriately to warnings issued. In addition, DPIE should continue to use best practice in the communication of air quality, such as the use of an air quality index (AQI), communication tools, and methods of disseminating the information efficiently, such as providing AQI information with radio, print, and electronic dissemination of daily weather forecasts [183]. Research in China has demonstrated the effectiveness of an AQI-like tool in communicating morbidity risks [184]. The design of this tool is important as the personal perception of air quality has been shown to have a stronger impact in modifying behaviour than a published index [185]. NSW already has an AQI and reports region-specific values [35], which is important due to the high spatial variability of air quality. This information is disseminated via websites, media outlets, email alerts, and mobile-phone alerts. Further resources could be provided to disseminate information on the NSW AQI and the public alert system as widely as possible, as well as evaluating the current system. Proper resourcing of the air quality monitoring and modelling efforts by DEPI needs to continue to ensure the best public health outcomes are achieved.

*3.6. Public Outreach, Education, and Community or Individual Actions Designed to Reduce Exposure*

Specific education campaigns that highlight the risks to health, the local sources, and the actions communities can take to improve air quality can also reduce overall exposure to air pollution.

Specific examples include:

1. Implementing strategies to encourage cycling and walking. Provide services such as safe cycling maps, bike lockers, and showers to encourage students and staff to walk or cycle to school. Cycling and walking do not contribute to poor air quality like many of the other modes of transport and offer the co-benefit of physical activity. In Sydney, the use of cars to travel to school is associated predominantly with the attitudes of parents, highlighting the need for an integrated (child-parent) approach in education strategies [186].

2. Support outreach and incentive programs to motivate the public to move away from the use of wood and other combustion heaters. Community education has been shown to have a significant effect on reducing wood smoke emissions in Australia, (by reducing use of wood-heaters) using health risk as a motivational trigger [187].

3. Use outreach and education programs to highlight the risks of both ambient and indoor air pollutants. Studies in Australia [122,125,188] and overseas [189,190] have highlighted the need for public health education with respect to the health risks of indoor air quality, especially as Australians spend the majority of their time indoors. Indoor air quality is unregulated in Australia [191] and therefore, can be very poor. A study in Brisbane showed times of the day likely associated with cooking and commuting were the largest contributors to ultrafine particle exposure in children [192].

4. Provide advice for reducing individual exposure. This should include:

   - Taking steps to minimise exposure to air pollutants by: exercising away from main roads, or, if this is not possible, then exercising in the early evening when the boundary layer is higher (in preference to the morning) [131] and choosing alternate activities when air quality is forecast/measured to be hazardous.
   - The potential benefits of behaviour that can reduce personal exposure to particulate air pollution during hazardous air pollution events. There is only limited evidence that adopting behaviours to limit personal exposure to air pollutants is effective in reducing cardiopulmonary health risks [193]; however, evidence demonstrating positive effects include altering air conditioner settings [194] and wearing a personal respiratory mask [195]. Although more recent research suggests that face masks could raise pollution risks [196].

5. Introduce anti-idling zones, especially around at-risk populations such as child-care centres, schools, aged-care homes, and hospitals. On-road vehicle emissions contribute to student air pollutant exposure [161], and morning and evening peaks in exposure have been measured [192]. Anti-idling has been shown to be effective in improving air quality in circumstances where the drop-off and pick-up zone traffic is a major component of local air pollution mix, although this mitigation measure is not as effective where schools are located very close to major highways [161,197–200]. The anti-idling efforts must be accompanied by appropriate education since community and driver knowledge of health benefits has been shown to increase with education efforts [201].

*3.7. Priorities for Further Research*

Continued research into air quality is required, especially as the move away from fossil fuels will change the atmospheric chemistry and shift the priorities from where they are at present. In particular, we recommend:

- A thoroughly researched and detailed National Air Pollution Emission Inventory should be funded (that incorporates and extends the existing one), including gridded and time resolved

emissions where appropriate and uncertainty estimates, with resources provided for annual updates. A national emissions inventory is crucial for the prioritisation of targets for pollution reduction and for determining effective air quality management policy and predicting future air quality scenarios [202]. There is no national emissions inventory for Australia comparable to those present in the USA [203] or the United Kingdom [204]. The existing Australian National Pollutant Inventory does not capture domestic, area, or line emissions and is, therefore, incomplete [202].

- Research on future air quality, exposure, and associated health impacts taking into account changing energy/fuel use, climate, population growth, and development as well as urbanisation.
- Research into 'data fusion' across existing air quality networks and future sensor networks that could include hot spot measurements, satellite retrievals, and model outputs (including chemical transport and land use regression models) for a more comprehensive air quality and exposure mapping, etc. [205].
- Make the measurement of indoor air quality a research focus. Indoor air quality is unregulated [191] and harmful [206]. Little research has been completed on indoor air quality [122,125,191].
- Research on pollen speciation, distribution, and health impacts. Support the development of pollen emission methodologies within air quality models to protect populated or otherwise at-risk areas of NSW. A system is being developed for Victoria [207] and could be extended to other regions of Australia.
- Research on the effect of urban greening on air quality. This should include amelioration of particulate matter by vegetation as well as biogenic VOC emissions (in order to improve the agreement between measurements and models [120]). In addition, further research into the atmospheric chemistry that follows biogenic VOC emissions and leads to secondary organic aerosol formation and ozone production should be prioritised.

### 3.8. Concluding Remarks Regarding Policy Implications

In this section, we have listed a large number of policy options and targets that should be prioritised. Since modelling of the environmental economics and health impacts of all the different policy options is beyond the scope of this work, we have not attempted a full ranking of these priorities. We recommend the support of future efforts for the construction of such modelling tools for use in all Australian jurisdictions. Nevertheless, it is clear, from the research that apportions contributions from different pollution sources, that residential wood-smoke, smoke from wildfires and hazard reduction burns, traffic emissions and emissions from industry and power generation contribute significantly to population exposures in Sydney. We therefore recommend prioritising policy analysis of measures to significantly reduce population exposures to air emissions, with consideration given to reducing the use of wood-heaters, minimising smoke impacts from bushfires and moving away from combustion-engine based transport in the Sydney basin.

### 4. Summary and Conclusions

This review provides an overview of the findings of research undertaken by CAUL and its partners, summarising the papers in the special issue of *Atmosphere* on Air Quality in New South Wales, Australia, in the context of previous research. Some of the highlights include:

1.  Publicly driven research by CAUL has provided case-studies in Sydney that can be used to deliver clear and simple messages about air quality to the public, such as:

    - The DPIE network of air quality monitoring stations is likely to be fit for purpose, with respect to representing urban background pollutant concentrations in Sydney, (i.e., in areas that are not close to a local pollution source, such as major traffic thoroughfares).

- Roadside pollution levels (such as $PM_{2.5}$ concentrations) are likely to be significantly higher than non-road side locations, with hotspots at traffic junctions, bus-stops, and drop-off and pick up zones (e.g., at schools).
- Air quality improves rapidly with distance from main roads so that pedestrians and cyclists are advised to use side-streets whenever possible.
- Due to meteorology, roadside pollution is often significantly worse in the morning rush hour than the afternoon rush hour, such that cyclists and joggers can reduce their exposure by choosing to exercise in the afternoons.

2. Novel measurements have allowed us to better understand the role of $NH_3$ in the chemistry of aerosol formation in Sydney and to understand the complex chemical mix of toxins that are present in wood-smoke, whether from bushfires or domestic wood-heaters.

3. Studies of the amelioration of air pollution in NSW have shown the capacity of urban trees to remove fine particulate matter from the atmosphere, and have highlighted the even greater efficiency of mosses in this capacity.

4. A major air quality modelling comparison has enabled the operational air quality forecasting model used for Sydney to be benchmarked against international standards, thereby increasing confidence in the daily forecasts.

Sydney's future air quality can be improved by reducing pollution from traffic, residential wood-smoke, industry, and power generators, and minimising smoke impacts from hazard reduction burning. There are also possible gains to be had from providing greater amelioration by greening the city with native trees and mosses; however, further research is needed to assess the pros and cons of urban greening for air quality.

**Author Contributions:** Conceptualization C.P.-W. and P.R.; original draft preparation C.P.-W., J.S. and S.L.F.; review and editing—all authors.

**Funding:** This research was funded by Australia's National Environmental Science Program through the Clean Air and Urban Landscapes Hub and Discovery funding 'Tackling Atmospheric Chemistry Grand Challenges in the Southern Hemisphere' (DP160101598). RS was also funded by ARC Centre of Excellence for Climate Extremes (CE170100023). YZ: travel supported by the University of Wollongong (UOW) Vice Chancellors Visiting International Scholar Award (VISA), the University Global Partnership Network (UGPN), and the NC State Internationalization Seed Grant at North Carolina State University, USA.

**Acknowledgments:** Thanks are due to the DPIE who provided some funding support and access to air quality monitoring data, and to the NSW EPA who provided access to the NSW GMR air emissions inventory data. YZ acknowledges supercomputer support from Stampede and Stampede 2, provided as an Extreme Science and Engineering Discovery Environment (XSEDE) digital service by the Texas Advanced Computing Center (TACC), and on Yellowstone (ark:/85065/d7wd3xhc) provided by NCAR's Computational and Information Systems Laboratory, sponsored by the National Science Foundation. Sonya L. Fiddes is supported by the Australian Research Council (ARC) Centre of Excellence for Climate System Science (CE110001028) and the Australian Government Research Training Program Scholarship. A.H. is in receipt of an RTP scholarship from the Australian Government. R.P. visited Australia with the assistance of an Endeavour Fellowship, financed by the Australian Government and was hosted by Sharon Robinson from the School of Earth, Atmospheric and Life Sciences (SEALS), University of Wollongong.

**Conflicts of Interest:** The authors declare no conflict of interest.

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
