# Peer review of "A Clean Air Plan for Sydney: An Overview of the Special Issue on Air Quality in New South Wales"

_atmosphere, doi:10.3390/atmos10120774_

Round 1

Reviewer 1 Report

Congratulations for the review that provides an exceptionally thorough and balanced analysis of the air quality issues in the Sydney region. Section 1 is a norm for similar reviews and environmental situation reports. Here I would only emphasize some of its strenghts:

observes the fact that air pollution health effects are far from linear and relatively low air pollution leads to comparable health effects than higher concentrations; assesses the complex chemical interaction of natural and anthropogenic pollutants; considers the co-benefits and tradeoffs of air quality and climate goals; investigates the meteorological dynamics behind air pollution episodes, identifies sea breeze as a key phenomena for high ozone levels; assesses air quality, pollens, heat waves and urban comfort in an integrated approach; considers the transformation of urban structure and population when linking climate change to urban atmospheric environment changes; proposes behavioral and urban structure measures besides emission reduction; balances the pros and concs of urban vegetation.

In Section 1.7, confidence intervals should be given together with the health statistics results.

Section 2 is a more technical but important review on modeling efforts and performance.

In general, the economic and urban planning expertise is missing from Section 3. Instead of simply listing targets and some ideas of the possible measures, decision support should specify the environmental priorities to help policymakers schedule the environmental goals in their socio-economical constraints. 

I find Section 3.2 superficial compared to the overall quality of the paper. In the case of the electric vehicles, the grid impacts should not be underestimated; and policies should not only promote EVs, but discourage rapid and peak-time charging and work towards building a sustainable charging management. See e.g. https://doi.org/10.1016/j.enpol.2012.08.074 and https://doi.org/10.1016/j.jpowsour.2010.09.119. Also, there is a socio-economic tradeoff between traffic reduction (public transport) and EV promotion, and their optimal balance should also consider urban space constraints and land use. While this topic clearly points beyond the scope of the manuscript, a more sophisticated standpoint regarding EVs would be necessary.

In lines 1022-1023, please include a reference for pollens: "with the lowest emissions of biogenic VOC species and allergenic pollens"

In lines 1051-1052, please include a reference for mildew:  "to minimise the build-up of indoor air pollutants and mildew."

Please use subscript x in NOx throughout the paper.

typos: "Buenos Aries" and "not too over interpret"

Please consider my comments regarding Section 3 before publishing the paper. 

Reviewer 2 Report

General comment:

This is a novel and original endeavor to tackle the vastly challenging problem of poor air quality adverse effect in New South Wale in Australia. The voluminous paper gears much to layman terms to outreach to the general public as air pollution is a public problem. Rightfully so, and the authors are to be commended for the condor and transparency sharing the problem to arouse public awareness. In fact the initiative can become part of the solution as awareness will spur local action which is the most effective way to abate and reduce pollution(s).

However, the paper seemed to have a larger focus on health effect, mitigation measures/infrastructures and policy making than atmospheric environmental science --- which this journal was supposedly designed to promote. Having commented so, I feel it can be overridden with the intent of a special issue for a particular region – this case is New South Wales in Australia.

Specific comments:

The comments hereafter are largely related to my “narrowly biased” pre-conceived notion about “Atmospheric Science” emphasis of this prestigious journal. With the authors’ referencing the accompanied modeling and previous campaign data, I think it is acceptable to have this paper to complete the picture for societal benefits.

In section 2.1.1 The finishing paragraphs quoted degree(s) of photosynthetic reduction and increase in wax deposition. There was broad explanation of urban stress but without parameterization schemes or formulations to speculate what were the likely geo-physical/chemical reason(s) for the changes.

In Section 2.1.2

When the underestimation of NH3 emission from vehicular exhaust was quoted: was it only ammonium gas or included ammonia species (likely the gaseous species dominates)?

The effort to further understand data collected from previous campaigns was advantageous. It is great that many of these valuable data set and findings are to be included in this Special Journal Issue.

Section 3 for mitigation policies was really cutting-edge management skills – accolades to the authors.
